# How the ‘Aerobic/Anaerobic Glycolysis’ Meme Formed a ‘Habit of Mind’ Which Impedes Progress in the Field of Brain Energy Metabolism

**DOI:** 10.3390/ijms25031433

**Published:** 2024-01-24

**Authors:** Avital Schurr

**Affiliations:** Department of Anesthesiology and Perioperative Medicine, University of Louisville School of Medicine, Louisville, KY 40202, USA; avital.schurr@gmail.com

**Keywords:** aerobic/anaerobic glycolysis, energy metabolism, habit of mind, lactate, meme, oxidative phosphorylation, pyruvate

## Abstract

The division of glycolysis into two separate pathways, aerobic and anaerobic, depending on the presence or absence of oxygen, respectively, was formulated over eight decades ago. The former ends with pyruvate, while the latter ends with lactate. Today, this division is confusing and misleading as research over the past 35 years clearly has demonstrated that glycolysis ends with lactate not only in cancerous cells but also in healthy tissues and cells. The present essay offers a review of the history of said division and the more recent knowledge that has been gained about glycolysis and its end-product, lactate. Then, it presents arguments in an attempt to explain why separating glycolysis into aerobic and anaerobic pathways persists among scientists, clinicians and teachers alike, despite convincing evidence that such division is not only wrong scientifically but also hinders progress in the field of energy metabolism.

## 1. Who’s Who of Muscle Carbohydrate Metabolism

They were the biggest names in the field of carbohydrates and energy metabolism. They all utilized muscle tissue as they attempted to decipher the puzzle of how muscles convert the monosaccharide, glucose, to energy necessary for muscle contraction. They were Gustav Georg Embden, Otto Fritz Myerehoff and Jakub Karol Parnas. In 1940, with the help of many other scientists in the field, this trio was responsible for the elucidation of the very first biochemical pathway in history, namely, glycolysis. This pathway, as originally proposed, converts glucose, a six carbon molecule, into two three-carbon molecules of lactate.

Embden was nominated eight times, between 1923 and 1933, for the Nobel Prize but was never awarded it. He died seven years before the final glycolytic pathway was published. Meyerhoff was awarded the Nobel Prize in Physiology or Medicine in 1922. In 1938, he escaped Germany from the Nazi regime and went to the United States of America. Parnas was arrested by the Russian KGB during the Jewish Anti-Fascist Committee affair on 28 January 1949 and died in prison a day later, supposedly of heart attack, during his interrogation.

The first hurdle that the formulators of the glycolytic pathway faced was the fact that under normal aerobic conditions, muscles produce very little, if any, lactate. Since lactate was considered a waste product of anaerobic muscle glucose metabolism, they had to conclude that under aerobic conditions, glycolysis would produce mainly pyruvate, the precursor of lactate. That conclusion received support from the work of Hans Adolf Krebs, who received the Nobel Prize in Physiology or Medicine in 1953 for his work on the tricarboxylic acid (TCA) cycle also known as the Krebs cycle [1]. In their publication, Krebs and Johnson suggested that a carbohydrate derivative (pyruvic acid?) combined with oxaloacetic acid to form citric acid. Since pyruvate is the glycolytic product that precedes lactate, the formulators of glycolysis decided that under aerobic conditions, this is the monocarboxylate that enters the TCA cycle. From then on, the glycolytic pathway is subdivided into “aerobic” and “anaerobic” glycolysis, where the former ends after 10 enzymatic steps with pyruvate, while the latter ends after 11 enzymatic steps with lactate (Figure 1). The one additional step that differentiates between the two is the one that converts pyruvate to lactate, catalyzed by lactate dehydrogenase (LDH). Lactate had a reputation as a waste product of anaerobic glucose metabolism long before the glycolytic pathway had been formulated. Consequently, in the mid-to-late 1920s and the early 1930s, when British scientists, Eric G. Holmes, Barbara E. Holmes and Charles A. Ashford [2,3,4,5,6,7,8,9,10,11], working on carbohydrate metabolism in brain tissue noticed that accumulated lactate is quickly dissipated in the presence of oxygen, their obvious interpretation was that oxygen hastens the removal of this waste product. Unfortunately, this finding and most of their excellent published brain research was mainly ignored by the “muscle” researchers and was completely forgotten for three quarters of a century [12].

From its discovery in the late 18th century and until the mid-1980s, lactate was believed to be formed mainly during anaerobic conditions as a useless end-product of glycolysis at best and a poisonous substance at worse. Ever since lactate was shown to form in muscles under anaerobic conditions, exercise physiologists and athletic coaches around the world have been pushing the idea that it is the cause of anaerobic muscle pain, the removal of which is a must for muscle recovery.

## 2. Habit of Mind Fuels Resistance to and Disregard of Findings That May Debunk the Dogma

The notion that lactate is a useless anaerobic glycolytic product, the accumulation of which in muscles can only be avoided under aerobic conditions that lead instead to the production of the lactate precursor, pyruvate, has become the prevailing belief. Exercise physiologists, biochemists and neuroscientists who investigated energy metabolism and biology teachers around the world who taught the basics of that field all bought into this dogma. In essence, “lactate is bad, pyruvate is good” has become a mantra where there are two types of glycolysis, aerobic and anaerobic. While it is understandable how this notion was formulated when one considers the knowledge available during the first half of the twentieth century, the knowledge available today (see below) presents several dilemmas that many scientists have chosen to ignore or circumvent. This is due, most likely, to a ‘habit of mind’ [13,14], a mental–habitual barrier that many find very difficult to cross. This would best explain the reactions to the research by George A. Brooks on skeletal muscle lactate shuttle and utilization [15,16,17,18] and by Avital Schurr and his colleagues who demonstrated that aerobic utilization of lactate as the sole energy substrate supports neuronal function [19,20]. First, the lactate shuttle offered by Brooks, which implies that lactate enters into mitochondria to be converted to pyruvate, the substrate of the TCA cycle, requires the presence of lactate dehydrogenase (LDH) intramitochondrially, the enzyme that converts lactate to pyruvate and NAD^+^ (nicotinamide adenine dinucleotide) to NADH. Shortly after Brooks published his findings regarding mitochondrial oxidation of lactate [16,17,21,22], two papers were published simultaneously in the same issue of *The Journal of Physiology* [23,24] questioning the validity of the claim that lactate can be oxidized via mitochondrial LDH. Both papers claimed that muscle mitochondrial preparations prepared in the respective laboratories of the investigators contain only traces of LDH, preventing mitochondria from utilizing lactate to support the function of the TCA cycle, in contrast to their ability to do so with pyruvate. 

To realize how strong an influence ‘habit of mind’ has, one only needs to recall how successful the, now defunct, lactic acidosis hypothesis of ischemic brain damage had been throughout the 1980s and 1990s [25,26,27]. Even four decades after the elucidation of the glycolytic pathway, it was very easy to persuade a large contingency of scientists who studied possible mechanisms of hypoxic and ischemic brain damage that the culprit behind such damage is no other than the “usual suspect”, i.e., lactate. While skepticism is an inseparable part of the scientific experimentation, the best way to challenge new, unexpected scientific findings is to attempt to replicate them. Neither in the case of the Brooks group’s findings nor in the case of the Schurr group’s research results were such attempts made. Both Sahlin et al. [23] and Rasmussen et al. [24] chose to challenge the findings of the Brooks group by showing that their own mitochondrial preparation cannot utilize lactate as a substrate for the TCA cycle, since their preparation did not contain a significant amount of LDH necessary for such function. However, as early as 1984, the presence of LDH has been shown in an isolated mitochondrial fraction from rabbit skeletal muscle [28,29] and the Kline group demonstrated the presence of mitochondrial LDH in several rat tissues and organs [30,31]. Nevertheless, no skepticism was raised regarding these findings after they were published. The challenge to the presence of mitochondrial LDH came only when such presence suggested that lactate could be a mitochondrial oxidative substrate, an idea that threatened the status quo, where pyruvate is the end-product of glycolysis when oxygen is available. Similarly, challenges were raised when (lactic) acidosis was demonstrated to have a protective effect against hypoxic/ischemic neuronal damage [32], a finding confirmed two years later [33], because that effect did not agree with the aggravating effect suggested by the lactic acidosis hypothesis [27]. Moreover, not only is lactic acid harmless during hypoxia/ischemia and somewhat protective, unexpectedly, lactate was shown to support neuronal function as the sole energy substrate [19]. That discovery, similar to the one made by Brooks and his colleagues in skeletal muscle, prompted the skeptics to question its validity using two prongs: The first one questioned the ability of lactic acid to protect against hypoxic/ischemic neuronal damage when it is commonly known to be the real cause of delayed neuronal damage post-ischemia. Bo Siesjo, the father of the lactic acidosis hypothesis of delayed ischemic neuronal damage [27], initiated a campaign aimed at debunking the findings about lactic acidosis being harmless and possibly beneficial post-ischemia. Siesjo’s hypothesis was founded on a study by Myers and Yamaguchi [34] who administered glucose to monkeys just prior to the induction of cardiac arrest. These monkeys, in comparison to monkeys administered with saline, exhibited aggravated delayed neuronal damage. Naturally, Seisjo postulated that the extra glucose on board during cardiac arrest (anaerobiosis) leads to extra production of lactic acid and hence, the aggravated damage. That pre-ischemic hyperglycemia correlates with aggravated delayed neuronal damage has been demonstrated in stroke patients, and thus, it is the standard recommendation to lower high levels of blood glucose prior to brain and heart surgeries. Nevertheless, no one at the time considered the possibility that lactic acidosis might not be the culprit. Moreover, re-examination of the study by Myers and Yamaguchi [34] highlights several flaws in its design, statistical analysis and interpretation. The lactic acidosis hypothesis was challenged [35] and later it was shown that hyperglycemia-induced aggravated ischemic damage is due to increased release of corticosterone rather than lactic acidosis [36]. The second prong questioned the ability of a waste product (lactate) to replace glucose, the universal obligatory substrate of energy metabolism, and even if it could be a substrate for ATP production, it should not be more than a minor contributor, never to replace glucose [37,38]. Clearly, the skeptics perceived the ability of neurons to aerobically utilize lactate for ATP production as a threat to the dominion of glucose as the ultimate energy substrate in the brain and elsewhere. Although such a “threatening” claim was never made, it is clear that under specific conditions, lactate could replace glucose, such as during recovery from neuronal insults [20,21], where lactate is absolutely the obligatory aerobic energy substrate, not glucose, and the preferable energy substrate during neuronal activation [39]. Nevertheless, upon the publication of the astrocytic neuronal lactate shuttle (ANLS) hypothesis by Magistretti and Pellerin in 1994 [40], a great debate and much criticism ensued that which is still continuing today. The Dienel group has been at the forefront of protecting glucose’s reputation as the brain’s obligatory energy substrate. While attempting to debunk the ANLS hypothesis, Gerald Dienel himself summarized in great detail his group’s standing on the issues and controversies that afflicted the topic of brain energy metabolism [41,42]. As for the obligatory role of glucose in brain energy metabolism, no disagreements should exist. After all, there would be hardly any lactate without glucose. There are two issues that continue to feed this three-decade-long debate on the possible role of lactate in energy metabolism, issues that are related to each other. The first has to do with the original drawing of the glycolytic pathway and its partition into aerobic and anaerobic glycolysis. Accordingly, the aerobic pathway requires 10 enzymatic reactions, the last one which produces pyruvate, the presumptive substrate of the mitochondrial TCA cycle. The anaerobic pathway requires 11 enzymatic reactions, terminating with the production of lactate, the monocarboxylate that for many decades was believed to be a useless waste product. That aerobic glycolysis terminates with the production of pyruvate, a postulate made 83 years ago, is thermodynamically unfeasible, since the standard free-energy (ΔG^0^) change in pyruvate conversion to lactate by LDH is −6 kcal/mol, which means that this conversion should ensue whether oxygen is present or absent [43]. A more recent paper [44] provides a different angle to assess the LDH reaction. These authors highlight the misconceptions regarding thermodynamic and enzyme kinetic considerations that may have led many to accept that the presence of oxygen could, somehow, block the LDH reaction from reducing pyruvate to lactate. Nonetheless, biochemistry textbooks and hundreds of online presentations simply ignore this fact and do not provide any mechanistic explanation for how the eleventh reaction of glycolysis is somehow being arrested in the presence of oxygen. Consequently, the prevailing definition of the pathway is “Glycolysis is the metabolic pathway that converts glucose into pyruvate”. As textbooks have become more or less obsolete and online information is the main go-to source, one could be overwhelmed by the countless “up-to-date” presentations of the glycolytic pathway, none of which include or refer to any of the findings by the research groups of Brooks and Schurr and the multitude of studies supporting these findings [45,46,47,48,49,50,51,52,53,54,55]. Moreover, lactate produced by other body tissues can be transported to the brain both as a signaling molecule and as an oxidative energy substrate [56]. Consequently, this eight-decade-old impossible concept has survived and is so deeply ingrained in our minds that it has become a habit we find difficult to abandon [13,14]. The second issue, previously mentioned, is the reluctance by many to accept the possibility that the glycolytic pathway, whether under aerobic or anaerobic conditions, always ends with the production of lactate. To gain a sweeping acceptance of this fact will require changing the drawing of the schematics of the pathway in multiple textbooks and even more online presentations. Such a change must involve the persuasion of many scientists, teachers and students to abandon their habit of mind. This status quo has manifested itself not only in the terminology used by investigators in the field, where “aerobic glycolysis” now means glycolysis that proceeds to produce lactate in the presence of oxygen [57], but also when real measurements of cerebral metabolic rates of both glucose and oxygen are made and analyzed [43].

An alternative approach to comprehend the concept of ‘habit of mind’ has been offered even earlier by the ‘meme theory’ of Richard Dawkins [58], who coined the term ‘meme’ as a unit of cultural transmission. It is proposed here that the idea of two types of glycolysis, aerobic and anaerobic, is a meme, “a unit of imitation”, replicating itself from brain to brain. Dawkins himself explains the meme unit: “*If a scientist hears, or reads about, a good idea, he passes it to his colleagues and students. He mentions it in his articles and his lectures. If the idea catches on, it can be said to propagate itself, spreading from brain to brain... When you plant a fertile meme in my mind you literally parasitize my brain, turning it into a vehicle for the meme’s propagation in just a way that a virus may parasitize the genetic mechanism of a host cell”.* The mantra of two separate glycolytic pathways, aerobic and anaerobic, fits Dawkins’ meme description, as it took hold in the brains of countless scientists and non-scientists alike and therefore is responsible for their ‘habit of mind’. Even prior to the formulation of the glycolytic pathway eight decades ago, the ‘aerobic’ and ‘anaerobic’ meme already had ingrained itself deeply enough among the scientists of the day to affect the interpretation of their own experimental results, as the above-mentioned case of Holmes, Holmes and Ashford [2,3,4,5,6,7,8,9,10,11] clearly indicates. Obviously, the problem is not terminology alone. Beyond the confusion that said terminology produces whenever the topic of energy production via glucose hydrolysis is dealt with, it undoubtedly impedes both our better understanding of this utmost important biochemical process and, consequently, potential treatments and cures of ailments and disorders that originate from energy metabolic abnormalities. 

## 3. From [Anaerobic Glycolysis → Lactate] to [Aerobic Glycolysis → Lactate]

To understand both the meaning of the term ‘aerobic glycolysis’ and the confusion that it may spread among researchers and students alike, one must follow its history. Otto Warburg and colleagues [59,60] were the first to use the term ‘aerobic glycolysis’ to describe the odd behavior of tumor cells, where glucose uptake and lactate production are increased compared to non-cancerous cells, despite the presence of ample amount of oxygen. This metabolic anomaly was named “the Warburg effect”. Today, almost a century later, the term is freely used whenever an organ, a tissue or a cell consumes glucose without or with a smaller amount of oxygen than expected, despite sufficient amount of oxygen present. Although this phenomenon, considered to be typical for energy metabolism of cancerous cells, is not the focus of this monograph, it indicates glucose metabolism via glycolysis that ends with the production of lactate. It does not proceed according to the accepted dogma of aerobic glycolysis that begins with glucose and ends with pyruvate, the product that supposedly enters the mitochondrial TCA cycle and OXPHOS, the process that requires oxygen. This distinction was acceptable and clear as long as it separated cancerous cells from normal cells. Then, two studies from the research laboratory of Marus E. Raichle were published [61,62]. The first study showed a focal physiological uncoupling between cerebral blood flow (CBF) and oxidative metabolism in response to somatosensory stimulation in humans. The second study observed a non-oxidative glucose consumption during focal physiologic neural activity in humans. Incidentally, that study was published almost two months after Schurr and colleagues published their results in the same periodical on the ability of lactate to aerobically support neuronal function [19]. A non-oxidative glucose consumption by activated neurons was as tantalizing an idea as lactate being an oxidative substrate for such neurons. Thirty-five years later, these two findings are still being debated among investigators of brain energy metabolism everywhere. Non-oxidative glucose breakdown all of the way to its terminal product, lactate, despite the presence of oxygen, does not agree with the tenet on which aerobic glucose metabolism was founded, i.e., this monosaccharide hydrolysis supposes to terminate with pyruvate as its end-product, which then enters the mitochondrial TCA cycle (Figure 1). Such an occurrence, which betrays the established meme, makes one wonder how activated neurons manage to ignore the presence of both oxygen and mitochondria to allow the glycolytic pathway to proceed through its 11th step to produce lactate? Of course, red blood cells (RBCs) are known to do just that, despite the presence of ample oxygen they carry in them, but RBCs do not have mitochondria [63]. Is there really a cellular mechanism that determines if and when glycolysis ends with either pyruvate or lactate regardless of the presence of oxygen and mitochondria? Or maybe lactate is always the end-product of the glycolytic pathway, independent of oxygen and mitochondria [12]? ‘Aerobic glycolysis’, the consumption of glucose without an accompanied consumption of oxygen, has now been shown by numerous studies to occur in the brain. Many of these studies have utilized the imaging technique known as BOLD (blood oxygen level-dependent) fMRI (functional magnetic resonance imaging) signaling to measure oxygen consumption upon neural activation while also measuring glucose consumption. The reader could find specific information on the continually debated issues regarding the interpretation of data produced by the BOLD fMRI technique [64], as this particular topic is beyond the objective of the present monograph. Nevertheless, the BOLD signal supposedly measures RBCs’ deoxyhemoglobin level in blood vessels contained within the brain area scanned via MRI. Such suppositions imply that the tissue surrounding blood vessels (astrocytes, pericytes, neurons, etc.) is either completely devoid of oxygen or if not, then the consumption of that oxygen is unaccounted for. Moreover, the BOLD signal consistently lags behind the evoked neural function that purportedly consumes the oxygen to produce the necessary ATP for that function [64]. Nonetheless, as has been shown by Hu and Wilson over 25 years ago [65,66], oxygen is available for immediate consumption upon demand in vivo, as they directly measured tissue levels of glucose, oxygen and lactate while applying electrical stimulations to the perforant path of the rat hippocampus. A reanalysis of these results [39,43] clearly showed that oxygen was readily available and was utilized by the stimulated neural tissue along with glucose. Moreover, with each additional stimulation, an increase in lactate aerobic utilization was observed, accompanied by a noticeable decrease in glucose utilization. Considering the fact that the BOLD fMRI signal is delayed by ~500 ms after the onset of the stimulus [64], one could surmise that the observed reduction in deoxyhemoglobin, (the BOLD fMRI signal) or alternatively, the increase in oxygenation in response to the stimulus, arrives too late to supply the energy required for the neuronal function responding to the stimulus. Therefore, one may conclude that such reoxygenation response indicates a replenishment of oxygen that was already consumed and had been readily available in response to the stimulus. If this is the case, should we question the use of BOLD fMRI signaling in determining CMR_O2_ (cerebral metabolic rate of oxygen)? Furthermore, does a lack of BOLD fMRI response during neural stimulation indicate ‘aerobic glycolysis’ (glucose consumption not accompanied by oxygen consumption)?

## 4. New Hypothesis to Circumvent a Flawed Dogma?

The above mentioned issues and questions illustrate how a successful meme that forms a habit of mind, on the one hand, and the use of a not-yet-proven methodology that may lead one to a conclusion not necessarily supported by said methodology, on the other, could create further confusion and misconception. A recent publication by Theriault et al. [67] is a case in point. In their hypothesis paper, the authors attempted to explain why ‘aerobic glycolysis’, a 15-fold less efficient ATP producing pathway than mitochondrial OXPHOS, is a tradeoff choice that cells make. They call it the “efficiency tradeoff hypothesis” and argue that the BOLD signal increases indicate such brain metabolism choices, especially for neuronal signaling in thin, informationally efficient axons. This hypothesis relies on postulates and terminology that hinder our ability to formulate the process of cellular energy metabolism accurately. The authors divide brain cellular energy metabolism into three separate biochemical pathways based on their input/output of substrates and products, respectively. First is cellular respiration, comprising glycolysis + OXPHOS, where 1 glucose + 6 O_2_ produces ~32 ATP + 6 CO_2_ + 6 H_2_O. The other two are anaerobic and aerobic glycolysis, both of which use 1 glucose to produce 2 ATP + 2 lactates, the first in the absence and the second in the presence of oxygen. According to this hypothesis, in respiration, there are a total of 170 reactions that produce 32 ATP, or 1 ATP per 5.3 reactions, while both anaerobic and aerobic glycolysis produce 1 ATP per 11 reactions, and therefore, cellular respiration is more efficient than either of the glycolytic pathways. Left unexplained is the ability of cellular respiration (glycolysis + OXPHOS) to arrest glycolysis’ 11th reaction, namely, the conversion of pyruvate to lactate via lactate dehydrogenase (LDH). In essence, cellular respiration is portrayed exactly as it was originally proposed by the fathers of the glycolytic pathway over 80 years ago, where aerobic glycolysis terminates with pyruvate. The popular meme and the habit of mind it formed allowed Theriault et al. [67] to ignore the fact that, thermodynamically, glycolysis cannot stop at the production of pyruvate, as already mentioned earlier. Consequently, the terms “anaerobic glycolysis” and “aerobic glycolysis” are meaningless [57]. This would change if data could show that the presence of mitochondria in the cellular vicinity of where glycolysis takes place somehow halts the conversion of pyruvate to lactate. Nevertheless, the authors of the proposed ‘efficiency tradeoff hypothesis’ are clearly aware, by their own writing, that lactate supports mitochondrial OXPHOS. They do claim that the conversion of lactate to pyruvate that is necessary to achieve such support is not sufficient, though they do not provide any proof that this is the case or that such conversion does occurs extramitochondrially, since recent publications suggest that lactate is the real mitochondrial substrate of OXPHOS, where it is converted to pyruvate intramitochondrially [43,68]. Of course, where cellular respiration is concerned, the oxygen/glucose index (OGI) is an important indicator of the amounts of these two respiratory substrates utilized during either rest or during activity. The theoretical OGI for a full oxidative conversion of glucose to CO_2_ and H_2_O is 6.0. In practice, this index, when using several different methodologies, has always been found to be lower than 6. When aerobic glycolysis is considered to be the main source of ATP supply, one should expect an OGI = 0. The problem with measuring oxygen utilization concerns its location in the tissue under study. Most measurements are performed on blood-borne oxygen, whether by using an oxygen isotope, an expensive and time-consuming route, or by relying on changes in the level of oxygenated hemoglobin in RBCs. Changes in cerebral blood flow (CBF) are also used to indicate changes in oxygen utilization. However, all of these measurements are founded on the notion that RBCs in blood vessels, arteries, veins, arterioles and venules are the only tissue compartment to contain oxygen, such that any increase in deoxyhemoglobin and/or an increase in CBF indicates an increase in oxygen consumption. In other words, oxygen is not to be found anywhere else except in blood vessels. However, as has been indicated earlier, the above-mentioned measurements/responses are all delayed significantly compared to the measurement of glucose consumption performed by imaging techniques and especially compared to the physiological response to the stimulation that induces the consumption of glucose and oxygen. That oxygen is readily available for neurons upon stimulation was already discussed earlier, shown in [65], analyzed in [39] and reanalyzed in [43]. The changes in neuronal tissue oxygen concentrations in response to stimulation [66] would not be registered by measuring changes in the level of deoxyhemoglobin due to its delayed response. Moreover, even the direct measurements [65] and their analysis [43] of dips (consumption) and rises (replenishment) in oxygen levels in response to stimulations do not necessarily reflect the total amount of oxygen consumed and replenished, since it appears that oxygen is never in short supply (Figure 2).

Investigators should consider measuring CO_2_ production rather than O_2_ consumption to have a better grasp on cerebral metabolic rates and on which metabolic pathway is involved. Consequently, the reliance of the ‘efficiency tradeoff hypothesis’ on the BOLD fMRI signal is questionable at best, considering that this methodology does not necessarily reflect the oxygen consumption that occurs instantaneously upon stimulation, a point that was also discussed earlier. Considering that the stimulated neuronal tissue is actually saturated with enough oxygen to fulfill most, if not all, of the needs for oxidative metabolism of glucose, any measurement of delayed oxygen extraction from blood vessels cannot account for an accurate estimate of the amount of oxygen consumed by the stimulated tissue. The authors who proposed the ‘efficiency tradeoff hypothesis’ themselves recognized the multiple drawbacks that the BOLD signal could introduce when it is the method by which oxygen consumption is being calculated. As tempting and as promising as the BOLD fMRI technique is, one is hard-pressed to accept it as a major determinant in reaching the conclusion that ‘aerobic glycolysis’, i.e., glycolysis that ends with lactate production despite an abundance of oxygen, is really what is observed during the cited experiments and studies that support the efficiency tradeoff hypothesis. Again, from the hypothesis paper itself, lactate production during stimulation is used to argue in favor of ‘aerobic glycolysis’; however, if one is to consider that lactate is always the final product of glycolysis, including during respiration, and that it is produced upon stimulation at rates that much exceed the levels consumed by mitochondria for ATP production [39,43], it should not be surprising that lactate efflux has been traced [67].

## 5. Glycolysis Always Ends with Lactate, the Aerobic Substrate of the Mitochondrial OXPHOS

The insistence by a great number of biochemists, physiologists, neuroscientists and physicians to continue their research and teaching of energy metabolism and, especially, cerebral energy metabolism using a flawed narrative of these metabolic pathways impedes the advancement of their research and results in hypothesizing new hypotheses to overcome that narrative. It is more than reasonable to accept the fact that there are two separate energy metabolic pathways, glycolysis, which hydrolyzes glucose to 2 lactates + 2 ATP, and mitochondrial oxidative phosphorylation (OXPHOS) that converts 1 lactate to 3 CO_2_ + 3 H_2_O + ~15 ATP. Glycolysis operates in the cytosol and is oxygen-independent, while OXPHOS operates in the mitochondrion and is oxygen-dependent. A popular meme and a habit of mind, unfortunately, are still the main hurdles for such acceptance, resulting in a mind forced to come up with new hypotheses to explain the flawed narrative. In the case of the ‘efficiency tradeoff hypothesis’, its authors proposed three different energy metabolic pathways to explain the measurements of cerebral energy metabolism using unproven methodology that measures oxygen consumption in response to stimulation to reach the conclusion that the least efficient energy producing pathway is a choice that cells make. In 1940, the proposed glycolytic pathway of Embden, Myerehoff and Parnas was drawn with the understanding that not all of the necessary details to draw an accurate series of 11 enzymatic reactions are yet available. At the time, they lacked any knowledge of the existence of mitochondria and their intricate workings were non-existent. Much has been learned and added to our knowledge since then about glycolysis and OXPHOS and yet, for too many in the field of energy metabolism, not much has changed. Interestingly, the editors of *Science*, the periodical that published the paper titled “Lactate-Supported Synaptic Function in the Rat Hippocampal Slice Preparation” [19] considered this paper important enough to include a picture of hippocampal slices as the cover picture for that issue (Figure 3). Evidently, they did not allow for the habit of mind to question the validity and the importance of the findings described in that paper.

## 6. Conclusions

Considering the current scientific data, along with the historical reflections and arguments that this monograph has detailed, a new meme is offered, namely, “glycolysis”; a metabolic pathway comprised of eleven sequential enzymatic reactions that breaks down a molecule of hexose (D-glucose) to produce two triose molecules (L-lactate). In the process, two ATP molecules are consumed, four molecules of ATP are produced and two NADH coenzyme moieties are oxidized to NAD^+^. Glycolysis occurs in the cytosol and is not affected in any way by the presence or absence of oxygen or mitochondria. Consequently, the meme offered here carries no prefix, neither “aerobic” nor “anaerobic”, just “GLYCOLYSIS”. It is anticipated that the acceptance of this ‘simpler’ meme will simplify and clarify not only the sequence of the first elucidated metabolic pathway in history, but also our understanding of its role in cellular energy metabolism in both health and disease.

## Figures and Tables

**Figure 1 ijms-25-01433-f001:**
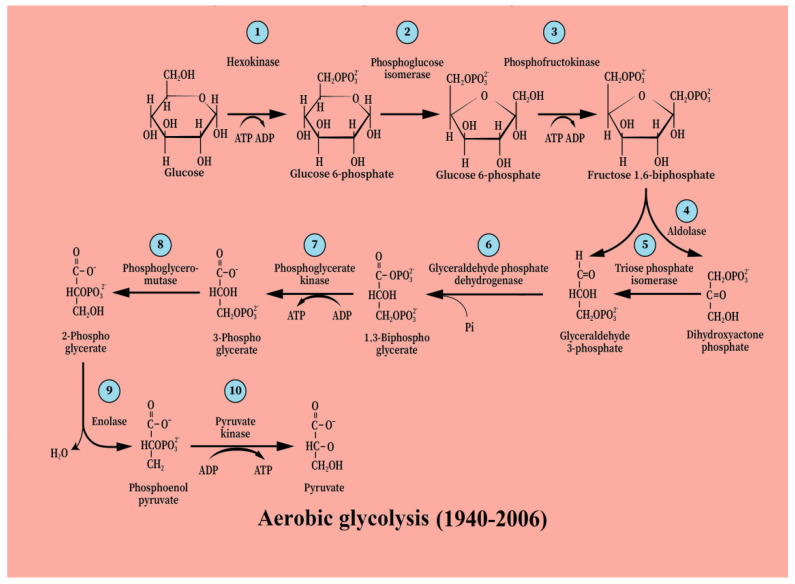
Glycolysis, the first biochemical pathway to be elucidated in 1940. It was divided into two separate pathways based on the presence (aerobic) or absence (anaerobic) of oxygen. It was determined then that aerobic glycolysis ends with the production of pyruvate, while anaerobic glycolysis ends with the production of lactate. Sixty-six years later, it was first suggested that glycolysis always ends with lactate regardless of the presence or absence of oxygen [12].

**Figure 2 ijms-25-01433-f002:**
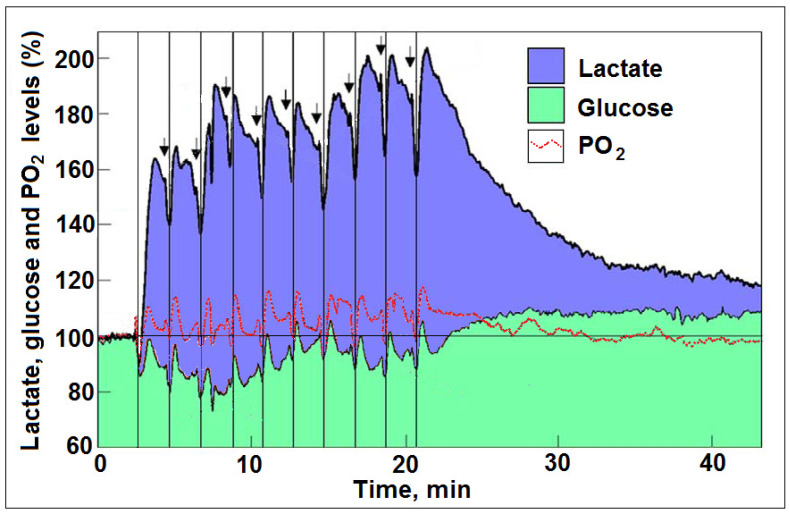
Profiles of time course and dynamic relationships between local extracellular L-lactate, glucose and O_2_ levels in the rat hippocampal dentate gyrus during a series of 5 s electrical stimulations (arrows) of the perforant pathway with 2 min rest intervals (reproduced from Hu and Wilson [65]). The changes in the mean concentration of glucose were always in the opposite direction to the changes in L-lactate concentration. The vertical lines are drawn to indicate the simultaneous dip in all three analytes in response to each of the electrical stimulations. Oxygen was clearly consumed with every stimulation along with both glucose and lactate and was never in short supply. In addition, with every stimulation, more lactate was produced and consumed than glucose. For additional details, see refs. [43,65].

**Figure 3 ijms-25-01433-f003:**
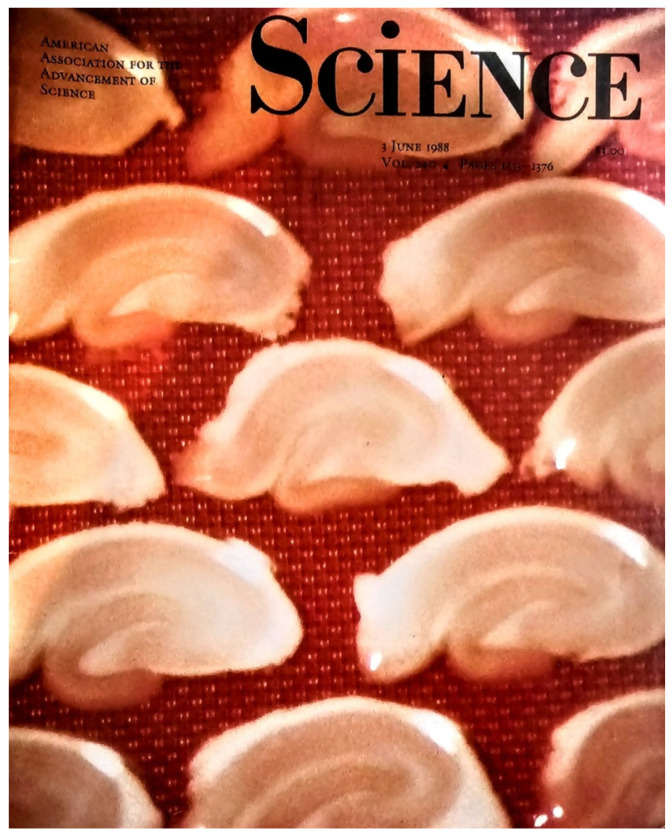
The cover picture of *Science* of the 3 June 1988 issue. Shown are rat brain hippocampal slices similar to those used in a study published in that issue [19]. The study demonstrated for the first time the ability of lactate, as the sole oxidative energy substrate, to support neuronal function. Habit of mind did not play a role in the decision made by the editors of *Science* at the time to publish this ground-breaking discovery by giving it the exposure they believed it deserved.

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
