# Peer review of "How the ‘Aerobic/Anaerobic Glycolysis’ Meme Formed a ‘Habit of Mind’ Which Impedes Progress in the Field of Brain Energy Metabolism"

_ijms, 2024, doi:10.3390/ijms25031433_

Round 1

Reviewer 1 Report

Comments and Suggestions for Authors

The manuscript, “How the 'Aerobic/Anaerobic Glycolysis' Meme Formed 'Habit of Mind' that Impedes Progress in the Field of Brain Energy Metabolism” by Avital Schurr is a detailed and argumentative essay addressing the pathophysiological role and molecular reactions of the metabolic pathway that converts glucose into pyruvate, called glycolysis. The author reviewed the history of research and recent knowledge about glycolysis and its end product lactate. The author convincingly argues why the division into two separate pathways, aerobic and anaerobic, depending on the presence or absence of oxygen, respectively, which have become widespread in recent years, has no scientific background and should be excluded from use by the scientific community. And I completely agree with this opinion. The manuscript is well written and may be accepted for publication in its current form.

Author Response

I would like to thank Reviewer 1 for his/her review and support. Some revisions have been made in the original manuscript based on the suggestions and comments of the other reviewers. thus, a graphic abstract had been added, the number of self-citations has been reduced. References added that support thermodynamic and enzyme kinetics considerations in favor of the conversion of pyruvate to lactate under any and all conditions. Also, a reference to a study that shows the presence of mitochondrial LDH, not only in brain and skeletal muscle, but also in liver and heart was added. 

Reviewer 2 Report

Comments and Suggestions for Authors

Cell metabolism is a complex process in which glycolysis, Krebs cycle and OXPHOS not only have the role to produce energy but also are involved in many catabolic and anabolic processes.

I am surprised about the subject of this manuscript in which the author tries to demonstrate that lactate is the end product of glycolysis and differentiates between aerobic and anaerobic glycolysis indicating that the accepted dogma about the use of these metabolic concepts is wrong.

I consider this manuscript does not offer scientific information apart of a strange discussion about the carbohydrate metabolism in cells.

It is known that cells show flexible metabolism depending on the growth that needs (mainly based on glycolysis and using TCA cycle as anabolic process) or quiescence that needs more oxidative metabolism involved in the generation of energy for cellular processes. In the case of neurons probably the use of glucose is not direct because glia cells can partially metabolize glucose to lactate and transfer lactate to neurons in which lactate is oxidized to pyruvate that enters in the mitochondrial metabolism to generate energy in form of ATP.

This does not necessarily means that the last molecule in glycolysis is always lactate. In fact, many metabolic diseases associated with mitochodrial dysfunction produce lactic acidosis because the survival of cells depends on the production of energy by glycolysis and the reduction of pyruvate to lactate restores the levels of NAD+ needed to drive glycolysis reactions.

For all these reasons I consider the manuscript very speculative, lacking the whole complexity of biochemical processes in cells that can be slightly different depending on the cell type or the activity as the physical activity in muscle.

Author Response

I thank Reviewer 2 for his/her comments. In addition to the references already included in the original manuscript, new references have been added to the revised draft in support of the main point i.e., that glycolysis always end up with the production of lactate, whether under aerobic or anaerobic conditions. In this respect, the separation of glycolysis into two pathways based on the presence or absence of oxygen is a wrong dogma. It is true that for lactate, the end-product of glycolysis, to be used as a mitochondrial substrate, oxygen must be available, but its glycolytic production is independent of oxygen availability. To offer additional support for the major point of this manuscript i.e., that glycolysis always ends with lactate, not pyruvate, the reviewer attention is directed to the reference by Rogatzki et al (2015), titled "Lactate is always the end product of glycolysis"  and an added reference to a paper by Bak and Schousboe (2017) that clearly explains why lactate dehydrogenase, the terminal enzyme of glycolysis, always catalyzes the conversion of pyruvate to lactate and how misconceptions about basic thermodynamics and enzyme kinetics led to erroneous conclusions in this regard. Accepting the fact that lactate is always the end-product of glycolysis should not have any bearing on any of the points the reviewer brought up:

"It is known that cells show flexible metabolism depending on the growth that needs (mainly based on glycolysis and using TCA cycle as anabolic process) or quiescence that needs more oxidative metabolism involved in the generation of energy for cellular processes. In the case of neurons probably the use of glucose is not direct because glia cells can partially metabolize glucose to lactate and transfer lactate to neurons in which lactate is oxidized to pyruvate that enters in the mitochondrial metabolism to generate energy in form of ATP."

That "many metabolic diseases associated with mitochondrial dysfunction produce lactic acidosis because the survival of cells depends on the production of energy by glycolysis and the reduction of pyruvate to lactate restores the levels of NAD+ needed to drive glycolysis reactions."                                             

After all, it has been established for some time now that lactate is an oxidative substrate for mitochondrial OXPHOS (see the added ref. by Young et al., 2020), that the ratio lactate/pyruvate both in brain and elsewhere is 15/1 or higher (see added ref. by Bak and Schousboe, 2017), that lactate enters mitochondria, where it is accumulating and converted to pyruvate before entering the TCA cycle (see ref. by Li et al., 2022) and that lactate produced by skeletal muscles is transported to the brain for oxidative consumption (see added ref. Bergersen, 2015). 

Overall, the manuscript is a historical examination of the scientific studies over the past century that have led to conclusions, not always supported by data, but rather based on assumptions, mainly due to lack of data. Upon the examination of the more recent studies and data, it has become clear that the older conclusions that formed the dogma, which separates glycolysis into two pathways, aerobic and anaerobic, were wrong. The scientific research of the past three decades strongly supports the concept of a glycolytic pathway that begins with glucose and ends with lactate, regardless of the presence or absence of oxygen. A graphic abstract has been added to the revised manuscript based on the suggestion of one of the reviewers and the added reference by Young et al (2020) supports a positive response to a question raised by Reviewer 3 on whether or not other tissues, beside brain and skeletal muscle, oxidize lactate mitochondrially.

Reviewer 3 Report

Comments and Suggestions for Authors

I really enjoyed reading this "sci-hi" (science-history) article. It is very well written and presents an accurate depiction of the history of glycolysis, from its discovery to our days, as well as a philosophical (so to speak) dissertation on how dogmas are often difficult surfaces to scratch, and extremely hard to die. 

My only suggestion would be, if possible, to draw a graphical abstract that summarizes the main differences between classical theories (1940-2006) and most current hypotheses.

Author Response

I thank Reviewer 3 for his/her comment and suggestion. A simple graphic abstract is included with the revised manuscript and several other changes, deletions and additions based on the comments by the other reviewers.

Reviewer 4 Report

Comments and Suggestions for Authors

The manuscript is devoted to the basic biochemical pathway 'glycolysis' and its traditional division into 'aerobic' and 'anaerobic'. Using mainly brain research data the author postulates the use of the only one term 'glycolysis' indicating the process of conversion of glucose to lactate which can be further used during OXPHOS.

The manuscript is marked as opinion and allow the readers to make a critical look on the current understanding of intracellular processes.

However, it is worth noting that metabolic pathways can vary significantly depending on the type of tissue. And this requires a similar analysis.

I think that the manuscript is well written and can be accepted after minor revision with the addition of data on the development of glycolysis in other tissues.

Author Response

I would like to thank Reviewer 4 for his/her review and comments. It is worth noting that much of the work on glycolysis and lactate production has been performed on skeletal muscle and brain tissues, as references by the Brooks' group show. Similar findings were reported with retinal tissue that could be considered part of the CNS. A reference was added to the manuscript showing the ability of mitochondria prepared from liver, heart and muscle tissues to support lactate oxidation (Young et al. 2020). Until additional research is performed on glycolysis in additional tissues, such as kidney, lung, etc., one could postulate that glucose hydrolysis that ends with lactate occurs in other tissues, as well. The revised manuscript includes several additions, deletions and changes suggested by the other reviewers.

Round 2

Reviewer 2 Report

Comments and Suggestions for Authors

The minor modifications introduced by the author only add some arguments to his main interpretation of the biochemical process of glucolysis. Metabolism is quite complex and this review only introduce an interpretation that indicates that cells use the reduction of pyruvate to lactate to further transfer lactate to the mitochondria to reoxidize it back to pyruvate that will be decarboxylated and enter into the mitochondrial ETC. In this process a mitochondrial lactate dehydrogenase can help increasing NADH into the mitochondrial matrix that will be used by mitochondrial OXPHOS complex I.

Although I consider the review quite speculative, it can add an interesting discussion in the biochemistry field.